# Assessment of Exposure to Time-Varying Magnetic Fields in Magnetic Resonance Environments Using Pocket Dosimeters

Giuseppe Acri [1] , Carmelo Anfuso [2], Giuseppe Vermiglio [3] and Valentina Hartwig [4],*

1    Dipartimento di BIOMORF, Università degli Studi di Messina, 98122 Messina, Italy
2    IRCCS Centro Neurolesi Messina, 98124 Messina, Italy
3    Independent Researcher, 98126 Messina, Italy
4    Institute of Clinical Physiology (IFC), Italian National Research Council (CNR), 56124 Pisa, Italy
*    Correspondence: valeh@ifc.cnr.it

**Abstract:** Staff working in Magnetic Resonance environments are mainly exposed to the static and spatially heterogeneous magnetic field. Moreover, workers movements in such environments give slowly time-varying magnetic field that reflects in an induced electric field in conductive bodies, such as human bodies. It is very important to have a practice method to personal exposure assessment, also to create a list of procedures and job descriptions at highest risk of exposure, to provide complete information for the workers. This is important especially for the "workers at particular risk", such as pregnant workers or medical devices wearers. The purpose of this work is to measure the exposure of the staff to time-varying magnetic field in Magnetic Resonance clinical environments, using pocket dosimeter. We present here the assessment of exposure in two different working conditions relative to routine procedures for different kinds of workers. The obtained results show compliance with the safety limits imposed by regulation for controlled exposure conditions. However, during the activity of replacement of the oxygen sensor performed by a maintenance technician, some exposure parameters exceeded the limits, suggesting to pay attention with specific conditions to prevent vertigo or other sensory effects.

**Keywords:** Magnetic Resonance Imaging (MRI); occupational exposure; exposure assessment; electromagnetic fields; personal dosimetry

## 1. Introduction

Magnetic Resonance Imaging (MRI) is widely used as a diagnostic non-invasive technique. MRI scanners are generally considered safe because they do not use ionizing radiation. Instead, MRI scanners use three different magnetic fields, including the static magnetic field ($B_0$), the radiofrequency (RF) field ($B_1$) and the magnetic field gradients, in the three spatial directions [1]. Literature studies regarding the interaction between electromagnetic fields in MRI environments and the human body have suggested that acute effects suffered by the staff are associated with the combination of time-varying magnetic field (dB/dt) and the strength of $B_0$ [2,3]. Hence, it is necessary to guarantee the safety of all the involved subjects that work within the MRI site (medical and technical staff). Due to this concern, International Commission on Non-Ionizing Radiation Protection (ICNIRP) published several guidelines which set exposure limits [3,4]. Guidelines have also been reported by the directive [5–7] issued by the European Parliament and the Council of the European Union.

In particular, the ICNIRP guidelines [4] show that the movement-induced electric field can evoke vertigo and other sensory perceptions such as nausea, visual sensations, known as magnetophosphenes and a metallic taste if the field intensity is high enough. There is also the possibility of acute neurocognitive effects, with subtle changes in attention, concentration and visuospatial orientation [8–11]. These effects are not considered to be hazardous per se, but they can be disturbing and may impair working ability.

The international guidelines pose safety limits to protect against short-term effects, such as stimulation of peripheral nerves and muscles, shocks and burns caused by touching conducting objects and elevated tissue temperatures resulting from absorption of RF energy during exposure to the electromagnetic field (EMF).

Only a few studies in the literature have reported chronic and long-term effects due to exposure to magnetic fields [12–15], but the confirmation of these findings is still lacking.

To verify compliance with exposure limits set by the regulation in force, as well as to characterize possible exposure scenarios in the framework of epidemiological studies on MRI occupational exposure, exposure assessment is highly recommended. However, at the moment, standardized procedures and methodologies for exposure assessment in MRI environments cannot be found despite extensive literature on the subject [16–18].

MRI staff are mainly exposed to the static and spatially heterogeneous magnetic field (fringe field). According to the Faraday's law, workers' movements in the fringe field also induce an electric field in electrically conductive bodies [19,20]. Hence, it is important to have a practice method to conduct personal measurements, to create a list of procedures and job descriptions at the highest risk of exposure and to provide complete information for the workers [21]. This is especially important for the "workers at particular risk," such as pregnant workers or those wearing active medical devices. In this last case, the regulation specifies a safety limit (action level, AL) of 0.5 mT to keep low interference with the function of active implantable medical devices (AIMDs). The regulation also sets an AL of 3 mT set to avoid the "projectile-risk" in the fringe field from strong sources (>100 mT) [5–7,22]; in this context, a prospective study conducted on an NMR spectrometer revealed that during workers' movement, already at a distance of 1 m from the spectrometer, employees were exposed to a static magnetic field >0.5 mT [23].

In the literature, recording systems, known as "dosimeters," have been used. Dosimeters are generally worn in workers' pockets and measure the instantaneous value of B [24–29]. The new-generation personal dosimeter [30] can evaluate the exposure to motion-induced TvMF during daily clinical practice [31–33].

The purpose of this work is to measure occupational exposure to time-varying magnetic fields in two MRI clinical environments using pocket dosimeters during an extraordinary maintenance task and a researcher activity.

## 2. Materials and Methods

Two exposure recording systems were used to assess the personal exposure of different operators (one maintenance technician and one researcher) in two different clinical environments, with a 3T MR scanner. The first dosimeter is a commercial device [24], while the second one is a new generation personal dosimeter [30].

Each dosimeter was fixed to the belt of the worker to assess the exposure of the torso area.

The procedure performed by the maintenance technician was an extraordinary maintenance task for the replacement of oxygen sensor. The task required the operator to stand over a ladder to work beyond the scanner (Figure 1a).

The task performed by the researcher was a non-standard procedure that required the presence of the operator in the scanner room during the examination. In this case, the operator performed a tactile stimulation protocol on the subject under a functional MRI (fMRI) session. The protocol required the operator to touch the subject's leg at set intervals. The operator stood close to the gantry for the entire duration of the examination (Figure 1b) and leaned out to touch the subject when necessary.

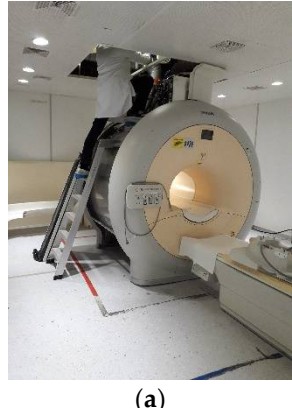
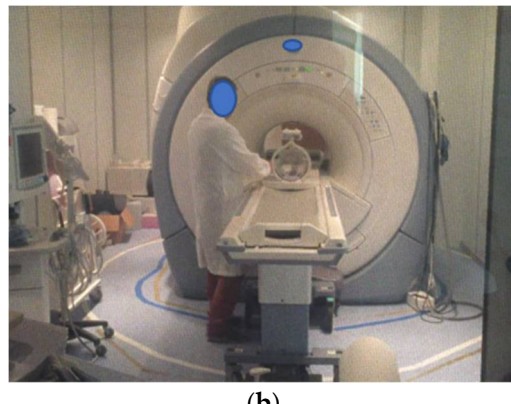

(**a**)               (**b**)

**Figure 1.** (**a**) Scanner #1: Philips 3T, worker: maintenance technician, task: oxygen sensor replacement. (**b**) Scanner #2: GE 3T, worker: researcher (engineer), task: tactile stimulation during fMRI protocol.

The values of the three spatial Bx, By and Bz components of the static magnetic field were analyzed using a homemade Matlab script to calculate the magnitude of the static magnetic field |B|. The raw data were downloaded and evaluated offline at the end of each task to obtain the magnetic flux density time gradient (dB/dt) and the induced electric field (EI) using the equation described in the ICNIRP guidelines [3]. As geometric factor C in the model, we chose 0.16 V/m per Tesla per second [19]. Hence, the results reported here are relative to an elliptical body cross-section of the human torso perpendicular to the magnetic field considering a person standing close to the scanner bore.

We also calculated the change of magnetic flux density during any 3 s period, ΔB, according to the ICNIRP recommendations.

Finally, the frequency spectra of the induced electric field and dB/dt were calculated to analyze the multiple spectral components. The waveform of the motion-induced electric field is a non-sinusoidal transient, hence the basic restriction of the induced electric field should be based on the weighted peak (WP) approach [3,4], as well as for the compliance verification with the reference levels for dB/dt. For this reason, the WP indexes were also evaluated according to the equation reported in ICNIRP guidelines [4] to verify compliance with the regulations (basic restrictions for induced electric field and reference levels for magnetic flux density). We considered exposure restrictions given by the authors of [3] for frequencies below 1 Hz and those given by the authors of [4] for frequencies above 1 Hz (upon conversion of BRMS into peak dB/dt in the case of reference levels).

## 3. Results

The peak value for each estimated exposure parameter ($B_{peak\ to\ peak}$, $EI_{peak}$, $dB/dt_{peak}$ and ΔB) is reported in Table 1 and compared with the exposure limits for the electric fields induced by movement in a static magnetic field and by time-varying magnetic fields below 1 Hz [3]. In addition, the WP index for basic restriction ($WP_{BR}$) and WP index for action level ($WP_{AL}$) are reported.

**Table 1.** Peak value for calculated exposure parameters. Red values represent the exceeded limits.

| | $B_{peak\ to\ peak}$ (T) | $EI_{peak}$ (V/m) | $dB/dt_{peak}$ (T/s) | ΔB (T) | $WP_{BR}$ | $WP_{AL}$ |
|---|---|---|---|---|---|---|
| **Maintenance task (technician)** | 2.430 | 0.263 | 1.646 | 2.524 | 0.193 | 0.514 |
| **fMRI task (researcher)** | 0.416 | 0.027 | 0.171 | 0.337 | 0.009 | 0.020 |

It is possible to observe that, for both the tasks performed by the researcher and the maintenance technician, none of the calculated parameters exceeded the limits imposed by the regulations for controlled exposure conditions. However, ICNIRP guidelines also set limits for $\Delta B$ (2 T) and $B_{peak\ to\ peak}$ (2 T) for uncontrolled exposure conditions specifically to avoid vertigo due to movement in a static B field or exposure to a time-varying B field. For the maintenance technician task, these limits ere exceeded.

Figure 2 shows the values of magnetic flux density B and dB/dt during the task of the maintenance technician (Figure 2a) and the researcher (Figure 2b). Figure 2a shows that, during the 10 min duration of the task, the worker exposure approached the limit of 2T several times until exceeding it (B = 2.430 T @Time = 466 s). Figure 2b, on the other hand, shows that the researcher was exposed to lower values of magnetic flux density during the entire length of the task (about 40 min) despite the proximity to the bore entrance. In this figure, the change in the magnetic flux density, at which the operator was exposed during the protocol that required to touch the patient under fMRI examination every 30 s for each acquisition sequence, is visible.

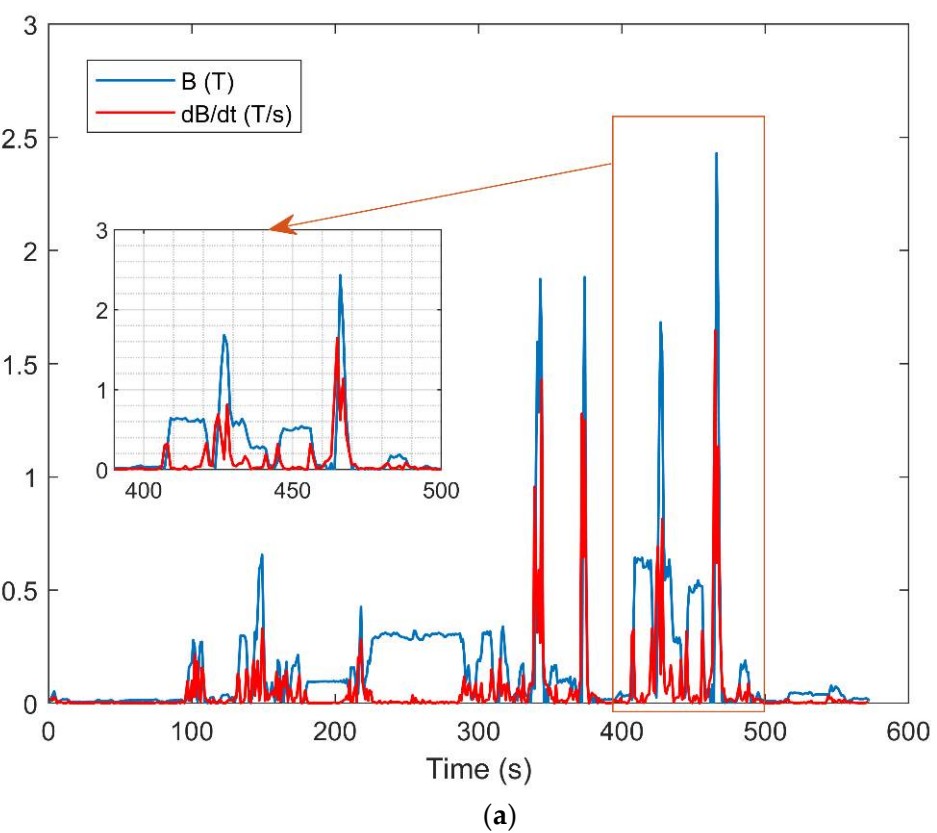

(**a**)

**Figure 2.** *Cont*.

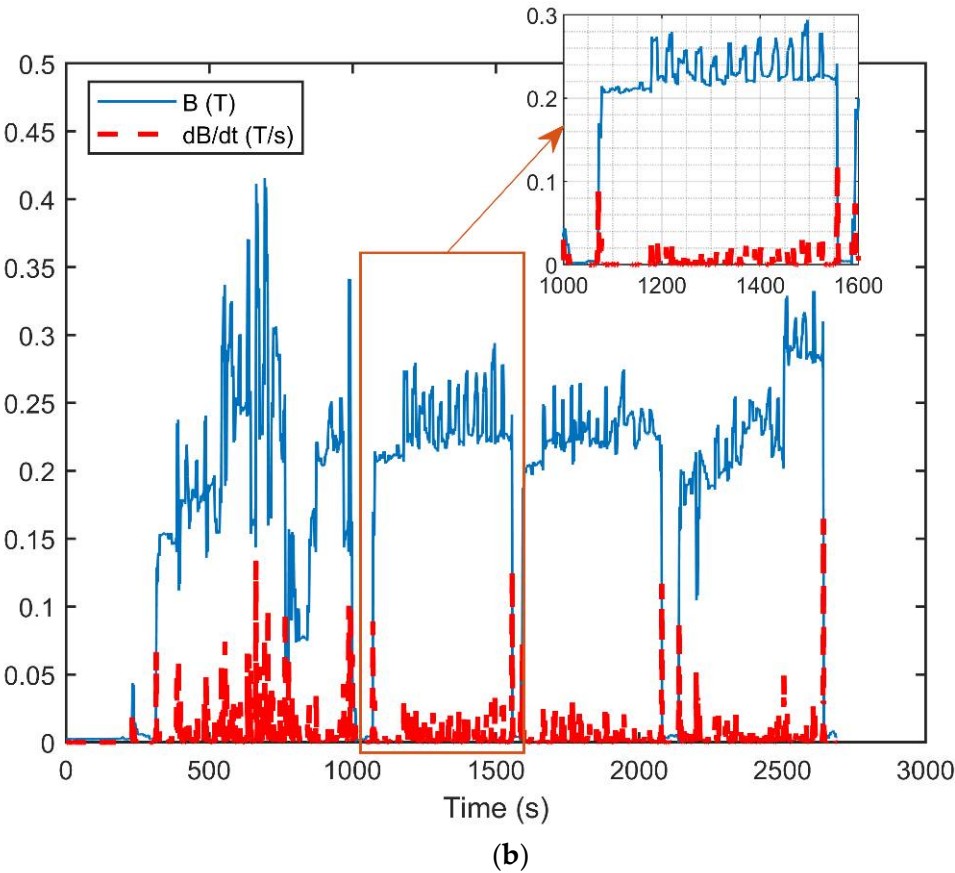

**(b)**

**Figure 2.** Magnetic flux density B together with its time derivative dB/dt experienced by: (**a**) a maintenance technician during oxygen sensor replacement, (**b**) a researcher during fMRI protocol.

## 4. Discussion

Accurate and reliable workers' exposure assessment is of great importance for risk assessment in the MRI environment. It is also considered an urgent research need, especially in view of epidemiological studies [34]. In recent years, the rapid evolution of MRI technology has increased awareness of the need to monitor the exposure conditions not just to verify the compliance of the exposure parameters with the safety limits but also to train the workers themselves to avoid possible risk situations. At the moment, standardized procedures and methodologies for the occupational exposure assessment in MRI environments cannot be found. Due to the highly variability of the exposure conditions, risk assessment in this environment is not a trivial task. Exposure assessment using a personal measurement system, such as dosimeters, can be useful to obtain a better characterization of the exposure conditions and to identify the factors determining such exposure variability.

In this work, we showed different exposure conditions of two different categories of workers in MRI using two personal dosimeters: a commercial dosimeter and an experimental one designed specifically for this purpose. Our results agreed with those reported in some literature studies where personal monitoring was carried out using portable dosimeters [32,35] or commercial three-axis Hall magnetometer [36,37]. First, the frequency spectra for both considered tasks showed that most of the frequency content was solved below 1 Hz. Considering the safety limits imposed by ICNIRP guidelines for controlled conditions relative to work environments where the workers are able to control their movements in order to prevent annoying and disturbing sensory effects, none of the calculated parameters exceeded the limits. However, if we consider the basic restrictions for uncontrolled conditions, which have been set in order to prevent transient sensory effects such as vertigo and nausea arising from motion-induced electric field below a few Hz [38], ICNIRP

recommends that the change of the magnetic flux density $\Delta B$ should not exceed 2 T during any 3 s period.

Our results show that this limit can be easily exceeded during the oxygen sensor replacement task performed by a maintenance technician, although the technician was trained on this possibility. This condition could be dangerous for the workers, considering that technicians perform this kind of task while standing on a ladder at a certain height above the floor. Therefore, the occurrence of vertigo could cause a fall. The maintenance technician was exposed to a higher magnetic flux density with respect to the researcher, likely because there was less shielding on top of the scanner. Moreover, the technician probably moved quickly to perform the assigned task, and this reflected in a high value of the change of magnetic flux density during the movement, i.e., the time integral of dB/dt. On the other hand, during the stimulation task, the researcher, was exposed to a lower magnetic flux density and probably moved slowly. Our results, which are relative to a research procedure requiring the operator to stand in the room while experiments such as fMRI are performed, show that the researcher was not exposed to any dangers. Indeed, it appears from our investigation that, despite the researcher staying near the bore, the exposure during scanning did not exceed the limits. In any case, exposures to RF and imaging gradients [19,39], and hazards from acoustic noise [2,40] were not evaluated in this work.

Although both workers were trained and used to work in the MRI environment, the maintenance technician was not sufficiently watchful to control his movements to prevent annoying and disturbing sensory effects.

This study has some limitations. First, the position of the dosimeter (fixed to the operator belt) was chosen to not interfere with the participants' work. After seeking the opinion of the workers themselves, we chose not to place the dosimeter on the head because it would have been inconvenient. To consider effects such as vertigo or dizziness, the assessment of exposure parameters on the head would be more appropriate [41]. However, as for X-ray dosimeters which are primarily worn on the chest, this would be the best placement in MRI settings for routine measurements.

Moreover, we chosen a geometric factor C relative to a generic elliptical body cross-section of the human torso without considering the specific size and shape of the operator. This geometric factor should also be calculated considering the position of the dosimeter for an accurate estimation of the induced electric field.

Future studies will focus on the monitoring of different exposure conditions considering further points of measurement. Moreover, any occurrence of exposure-related symptoms will be noted to associate sensory effects with specific working conditions and behaviors.

## 5. Conclusions

In this study, the time-varying magnetic field exposure was evaluated using pocket dosimeters worn by a maintenance technician and a researcher. The measurements were conducted on two different 3.0 T scanners. The obtained results showed compliance with the safety limits imposed by regulation. However, during the maintenance technician activity to replace the oxygen sensor, the $\Delta B$ exceeded the limit set to prevent vertigo due to movement in static field, which can be dangerous in specific conditions.

Once again, it was shown that personal exposure assessment can be useful to identify preventive measures and to provide staff with recommendations on how to move around an MRI room during the daily practice. This study also explores the great variability of exposure conditions in such environments.

## 6. Patents

The homemade personal dosimeter [30] used to acquire the magnetic flux density experienced by the researcher is the subject of a national patent (IT # 102017000114889).

**Author Contributions:** Conceptualization, G.A., C.A. and V.H.; methodology, V.H. and G.A.; software, V.H.; formal analysis, V.H. and G.A.; investigation, C.A.; resources, G.A.; data curation, V.H. and G.A.; writing—original draft preparation, G.A. and V.H.; writing—review and editing, V.H., G.A. and G.V.; visualization, V.H.; supervision, V.H. and G.V. All authors have read and agreed to the published version of the manuscript.

**Funding:** This research received no external funding.

**Conflicts of Interest:** The authors declare no conflict of interest.

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
