# Peer review of "Assessment of Exposure to Time-Varying Magnetic Fields in Magnetic Resonance Environments Using Pocket Dosimeters"

_electronics, doi:10.3390/electronics11172796_

Round 1
Reviewer 1 Report
The authors present their study on MRI dosimeters to measure the exposure of MRI workers. The work is interesting and also of importance in the daily clinical routine. The manuscript is well written, the methods are appropriate. The results are interesting, given that the worker had a higher exposure, but it seems logical since there is less shielding on top of the scanner, because nobody is supposed to be up there.
I have some minor comments:
1. Language. The authors need to rework the manuscript, there are several mistakes especially regarding grammar, e.g. "MRI scanners are generally considered safe because don’t use ionizing radiation" should read "because they don't use". These things are in the entire manuscript, so please have this checked specifically for the sake of readability.
2. Introduction: You claim "workers movements in such environments give slowly time-varying magnetic field that reflects in an induced electric field in conductive bodies,". This seems to me as if the bodies emit the magnetic field. Or do you mean that the movement disturbs the main magnetic field? Then please rephrase.
3. Results/Discussion: It is worthwhile elaborating more on the researchers' exposure during scanning. You mention it in the results, but it should be discussed as well, especially regarding the safety issue and that standing in the room while experiments such as EPI sequences are acquired do not expose any dangers (except for acoustic noise of course)
4. Limitations: You mention the dosimeters on the body, not on the head. While it is acadademically correct, this is no resoning for routine measurments. Hence, X-Ray dosimenters are also worn on the chest mostly. This would be the same placement in MRI settings I suppose, so this limitation you mention can be weakened.
Author Response
Dear Reviewer 1,
Thank you very much for your comments. Following your remarks, we revised our manuscript and below you can find a point-by-point response to your criticisms and suggestions:
- The authors need to rework the manuscript, there are several mistakes especially regarding grammar, e.g. "MRI scanners are generally considered safe because don’t use ionizing radiation" should read "because they don't use". These things are in the entire manuscript, so please have this checked specifically for the sake of readability.
We carefully checked the entire manuscript and correct some language errors.
- Introduction: You claim "workers movements in such environments give slowly time-varying magnetic field that reflects in an induced electric field in conductive bodies,". This seems to me as if the bodies emit the magnetic field. Or do you mean that the movement disturbs the main magnetic field? Then please rephrase.
We changed the sentence as follow:
MRI staff are mainly exposed to the static and spatially heterogeneous magnetic field (fringe field). According to the Faraday’s law, workers’ movements in the fringe field induce also an electric field in electrically conductive bodies[19,20].
- Results/Discussion: It is worthwhile elaborating more on the researchers' exposure during scanning. You mention it in the results, but it should be discussed as well, especially regarding the safety issue and that standing in the room while experiments such as EPI sequences are acquired do not expose any dangers (except for acoustic noise of course)
We added, in the Discussion section, a brief paragraph to discuss the researcher’s exposure during scanning:
On the other hand, the researcher, during the stimulation task, was exposed to a lower magnetic flux density and probably moved slowly. Our results, which are relative to a research procedure requiring the operator to stand in the room while experiments such as fMRI are performed, show that the researcher is not exposed to any dangers. Indeed, it appears from our investigation that, despite the researcher staying in close proximity to the bore, the exposure during scanning does not exceed the limits. In any case, it is understood that exposures to RF and imaging gradients [19,39], and hazards from acoustic noise [2,40] were not evaluated in this work.
- Limitations: You mention the dosimeters on the body, not on the head. While it is acadademically correct, this is no resoning for routine measurments. Hence, X-Ray dosimenters are also worn on the chest mostly. This would be the same placement in MRI settings I suppose, so this limitation you mention can be weakened.
We thank the Reviewer for this comment. We added this sentence in the Discussion Section:
Wanting to consider effects such as vertigo or dizziness, the assessment of exposure parameters on the head would be more appropriate [41]. However, as for X-Ray dosimeters which are worn on the chest mostly, this would be the best placement in MRI settings for routine measurements.
Reviewer 2 Report
First of all, thank you very much for the material sent. The subject of the publication is desirable. Nevertheless, I have some suggestions:
1. The theory of the electromagnetic field was not presented.
2. The static distribution, critical values of the field in the room during the operation of the device, were not presented in rare software (for example FEM).
3. The methodology of measuring the value of the field around the device (selected points, their identification) was not presented.
4. Hence, the work is of an engineering nature, not a research one.
Best regards.
Hence, I kindly ask you to correct the text in this regard, even if the authors would spend more time.
Author Response
Dear Reviewer 2,
Thank you very much for your comments. Following your remarks, we revised our manuscript and below you can find a point-by-point response to your criticisms and suggestions:
- The theory of the electromagnetic field was not presented.
The purpose of this study is to measure the occupational exposure to time-varying magnetic field in two MRI clinical environments, using pocket dosimeter which is able to measure the instantaneous value of B. The obtained results are compared with the safety limits imposed by regulation. In this context, we think it is redundant to insert the theory of the electromagnetic field that are recalled in the references used in the Introduction Section
- The static distribution, critical values of the field in the room during the operation of the device, were not presented in rare software (FEM)
The static field distribution, the critical values of the field in the room are generally of interest of MRI manufacturers during the device development and implementation.
The aim of this study was to conduct a risk assessment for MRI workers by introducing a different device based on experimental measurements in order to evaluate the dB/dt peak values in routine practice. The availability and applicability of the proposed methodology has been tested on different MRI devices. The obtained results confirmed the flexibility of the method, its simplicity, and its reproducibility as an efficient tool to evaluate the dB/dt peak values. In addition, the obtained results were compared with the ICNIRP restrictions.
In the meantime, we propose that the MRI workers should adopt specific behaviors especially when they move near the MRI gantry.
- The methodology of measuring the value of the field around the device (selected points, their identification) was not presented.
Thank you for your comment. In this study we have tried to obtain “real” information about workers’ exposure during manufacturer technicians’ operations. For this reason, it is impossible to select points during the measurements. The technicians worn the pocket dosimeter and then they worked regularly, so all the acquired measurements are relative to the position of the pocket dosimeter on the volunteers (on the belt).
- Hence, the work is of an engineering nature, not a research one.
In this manuscript we used also a new device (pocket dosimeter) to evaluate the occupational exposure to time-varying magnetic field in two MRI clinical environments. In our opinion, the focus of our study perfectly matches the aims of the Journal and the Special Issue. Moreover, we present the manuscript as technical note.
Round 2
Reviewer 2 Report
Many thanks to the authors for their answer.Nevertheless, I maintain my opinion on the presented material.
Best regards.